

# Ethylene represses the promoting influence of cytokinin on cell division and expansion of cotyledons in etiolated *Arabidopsis thaliana* seedlings

Ekaterina Stoynova-Bakalova[1], Dimitar V. Bakalov[2] and
Tobias I. Baskin[3]

[1] Institute of Plant Physiology and Genetics, Bulgarian Academy of Sciences, Sofia, Bulgaria
[2] Department of Pathophysiology, Medical University of Sofia, Sofia, Bulgaria
[3] Biology Department, University of Massachusetts at Amherst, Amherst, MA,
United States of America

## ABSTRACT

The plant hormones ethylene and cytokinin influence many processes; sometimes they act cooperatively, other times antagonistically. To study their antagonistic interaction, we used the cotyledons of etiolated, intact seedlings of *Arabidopsis thaliana*. We focused on cell division and expansion, because both processes are quantified readily in paradermal sections. Here, we show that exogenous cytokinins modestly stimulate cell division and expansion in the cotyledon, with a phenyl-urea class compound exerting a larger effect than benzyl-adenine. Similarly, both processes were stimulated modestly when ethylene response was inhibited, either chemically with silver nitrate or genetically with the *eti5* ethylene-insensitive mutant. However, combining cytokinin treatment with ethylene insensitivity was synergistic, strongly stimulating both cell division and expansion. Evidently, ethylene represses the growth promoting influence of cytokinin, whether endogenous or applied. We suggest that the intact etiolated cotyledon offers a useful system to characterize how ethylene antagonizes cytokinin responsiveness.

## INTRODUCTION

In the development of plants, cell division and expansion are coordinated by hormones. For studies of hormone action, a popular experimental material is the *Arabidopsis thaliana* etiolated hypocotyl. Indeed, this material grows rapidly and senses its environment actively. However, in the etiolated hypocotyl, cell division is minimal if not absent entirely, precluding comparative studies on expansion and division. Another often-used material is the root of the same species. However, in the root, division happens in the meristem while rapid expansion happens in the elongation zone; this zonation itself is subject to an elaborate system of hormonal control (*Wachsman, Sparks & Benfey, 2015*). Thus in the root, hormones regulate not only the base processes of division and expansion, they also regulate the positions where these activities occur (*Baskin, 2013*).

Corresponding author
Tobias I. Baskin, baskin@umass.edu

For studying both cell growth and division, an advantageous organ is the epigeal cotyledon. Emerging above the soil, epigeal cotyledons expand and become photosynthetic. Despite their small size, epigeal cotyledons play a leading role in plant success and underlie heterosis (*Wang et al., 2019*). In epigeal cotyledons, cell division and expansion happen congruently. Cells in these cotyledons grow primarily in the plane of the lamina ("laterally") and divide primarily anticlinally; furthermore, when cells do divide, they form prominent and long-lived clusters, allowing cells from each successive round of division to be distinguished in an assay at a final timepoint. Based on these attributes, cell division and expansion in the cotyledon can be quantified accurately from paradermal sections (*Stoynova-Bakalova, 2007*).

In cotyledons as in leaves, a hormone long known to play a significant role in promoting growth is cytokinin (*Ikuma & Thimann, 1963*; *Letham, 1971*; *Stoynova-Bakalova & Petrov, 2006*; *Skalák et al., 2019*; *Wu et al., 2021*). The hormone stimulates expansion of cotyledons, through a mechanism that is apparently independent of acid growth (*Ross & Rayle, 1982*), and also stimulates cell division (*Chory et al., 1994*; *Martin et al., 1997*). The hormone's stimulatory effect on division is mediated at least in part *via* up-regulating a transcription factor in the basic helix-loop-helix class, named CYTOKININ-RESPONSIVE GROWTH FACTOR (*Park et al., 2021*).

Rather than promoting growth as in the cotyledon, cytokinin usually inhibits growth of both the hypocotyl and root (*Deikman, 1997*; *Li et al., 2021*). For these inhibitory responses, cytokinin was discovered to act by stimulating the synthesis of the hormone ethylene. For example, cytokinin was found to be largely unable to inhibit root or hypocotyl growth in mutants that are insensitive to ethylene (*Cary, Liu & Howell, 1995*). The mechanism linking the hormones was partly accounted for by cytokinin stabilizing one of the enzymes that synthesizes ethylene (*Vogel et al., 1998b*; *Hansen, Chae & Kieber, 2009*) and because response to both hormones involves so-called two-component regulators that share components (*Hass et al., 2004*; *Zdarska et al., 2015*; *Iqbal et al., 2017*; *Bidon et al., 2020*).

By contrast to cytokinin's inhibition of growth of roots and hypocotyls, the hormone's stimulation of growth in cotyledons has typically been considered to be independent of ethylene. Consistently, growth of excised cucumber cotyledons is largely insensitive to exogenous ethylene (*Green, 1983*). For etiolated, intact cotyledons, the promotion of growth by cytokinin was reported to be independent of ethylene in *A. thaliana* (*Cary, Liu & Howell, 1995*) and tobacco (*Genkov et al., 2003*), although neither paper assayed cotyledon growth quantitatively. In principle, cytokinin could inhibit ethylene synthesis in the cotyledon as opposed to promoting synthesis in roots and hypocotyls, but ethylene being a gas would be expected to diffuse rapidly from root and hypocotyl and reach the cotyledons.

Contrary to the above claims for cytokinin influencing cotyledon growth independently of ethylene, cytokinin simulated the expansion of intact, etiolated *A. thaliana* cotyledons to a greater extent when ethylene responsiveness was inhibited (*Cortleven et al., 2019*). Such an antagonistic relationship between the two hormones is similar to their relationship during leaf senescence (*Jan et al., 2019*). Nevertheless, *Cortleven et al. (2019)* also reported

that the inhibition of growth by cytokinin in both the root and hypocotyl is independent of ethylene, for both light- and dark-grown material. The latter findings contradict several reports for etiolated material (*Cary, Liu & Howell, 1995*; *Vogel et al., 1998a*) as well as a report for light-grown material (*Smets et al., 2005*). While the reasons for the discrepancies between these papers are not clear, they suggest that the role of ethylene in the stimulation of cotyledon growth by cytokinin cannot be considered settled.

Here, we sought to clarify the relationship between cytokinin and ethylene in regulating growth and division of the intact, etiolated cotyledons of *A. thaliana*. We show that when ethylene response is inhibited, cytokinin powerfully stimulates not only expansion but also cell division. These results suggest that in the etiolated cotyledon, the growth promoting effects of cytokinin are held in check, perhaps giving way only when the cotyledon is pushed out of the soil allowing cytokinin and light to work together (*Zdarska et al., 2015*; *Cortleven et al., 2019*).

## MATERIALS AND METHODS

### Plant material and growth conditions

Seeds of *Arabidopsis thaliana* L. (Heynh), Columbia background (referred to here as *wild type*) and those of *eti*5, derived from the same background (*Harpham et al., 1991*), were used. The seeds were surface-sterilized for 15 min in 1 mL of 20% (v/v) commercial bleach, rinsed three times with sterile water (1 mL each time) and plated in 10 cm-diameter Petri dishes containing 20 mL of agar (8 g $L^{-1}$) solidified in distilled water (control). When indicated, 20 μM $AgNO_3$ or 10 μM cytokinin, either N6-benzylaminopurine (BA) or N- (2-chloro-4-pyridyl)-N′-phenyl-urea (4PU-30) was added to the medium before autoclaving. After sowing, the seeds were stratified at 4 °C for 2 days in the dark and then exposed to white light (50 μmol m$^{-2}$ s$^{-1}$) for 3 h to ensure synchrony of germination. Seeds were then germinated in a growth chamber at 22 °C in complete darkness and grown for up to 10 days.

### Light microscopy

Cotyledons were fixed in 3% glutaraldehyde, 0.2 M phosphate buffer, pH 7.2, dehydrated, and embedded in Spurr's epoxy, as described previously (*Stoynova-Bakalova et al., 2004*). The cotyledons were flat embedded by positioning them on a fine nylon net and then pressed with a transparent piece of plastic during the embedding. Serial, paradermal sections (2 μm thick) cut parallel to the surface of the cotyledons were made. The sections were mounted on standard microscope slides, stained with 0.01% (w/v) toluidine blue and observed through a light microscope (Carl Zeiss, Jena, Germany).

### Determination of cell division, cell growth, and cotyledon growth

For cotyledon area, 20 cotyledons were measured per treatment. For cell division and cell expansion, the number of cells per cell cluster was assayed as was the cluster's area. A cluster is recognized as two or more cells surrounded by a thick, continuous usually thicker cell wall (Figs. 1A, 1B). Cell clusters were assayed (area, cell number) only from the central part of the blade, defined as the region enclosed by the ring of vascular bundles (Figs. 1A, 1C).
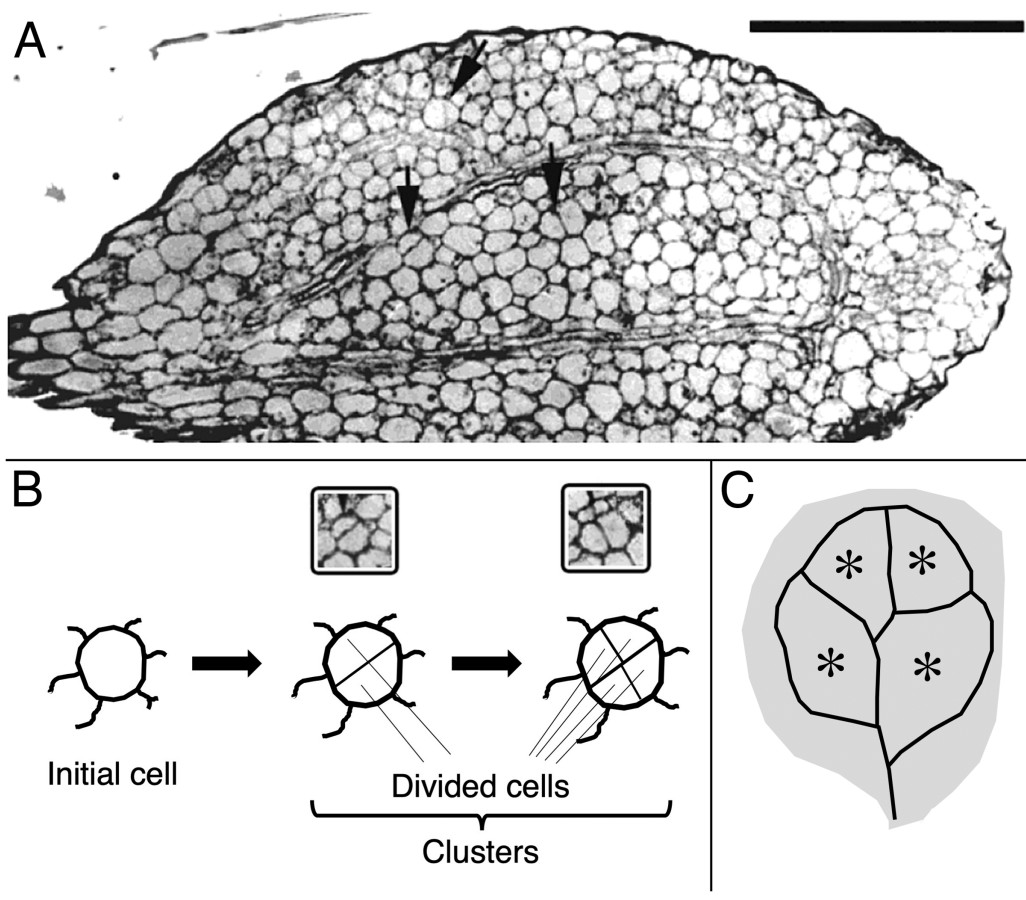

**Figure 1 Image and schematic showing features of cotyledon anatomy.** (A) Paradermal section through the spongy mesophyll of a wild-type cotyledon at the second day of development in darkness. Some cell divisions are visible (arrows). Bar = 100 μm. (B, C) Schematic representations of (B) cell clusters and (C) cotyledon regions. *Initial cell* refers to a cell present in the dry seed. For quantification, cell clusters were sampled only within the vascular ring (asterisks in the schematic). The section shown (A) curls out of the plane so that the vascular ring is not present in the lower half of the section.

For each treatment and tissue type, clusters were assayed among the serial sections, cut from 10 to 12 cotyledons fixed on three separate occasions. In the results presented for cell numbers and areas, data for at least 120 cell clusters are shown, pooled from all of the samples.

The microscope images of the sections were captured and saved on a digital image processor (International Micro-Vision Inc., Redwood City, CA, USA). The number of cells and area of cell clusters were measured with 3DDoctor software (Able Software Corp., Lexington, MA, USA). To measure cotyledon area, the seedlings were flattened under a coverslip and scanned. Area was measured on the images with the above software.

A comparison of means was performed by the Fisher LSD test ($P < 0.05$) after performing ANOVA-2 analysis (GraphPad Prism 8, https://www.graphpad.com) comparing cotyledon area, cell size, and the number of cells.
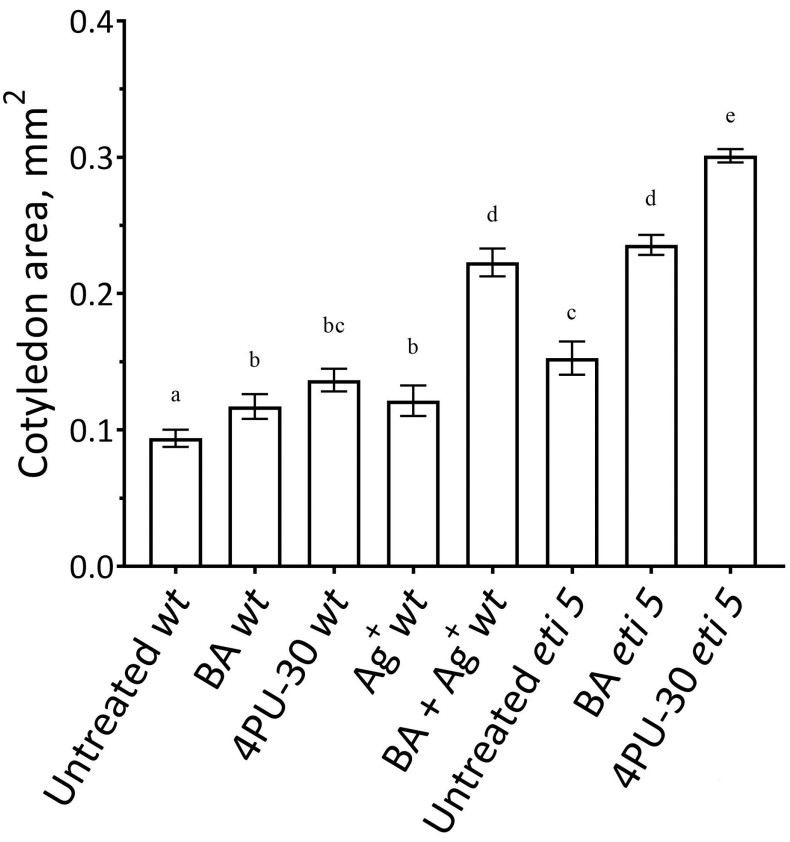

**Figure 2 Cotyledon area.** The area was measured after 10 days of growth in darkness. Values are means ± SE, $n$ = 20 per treatment group. Means in a column without a common superscript letter (a–e) differ ($P < 0.05$) as analyzed by two-way ANOVA and Fisher LSD test. In the treatment names, '*wt*' is wild type (Columbia); 'BA' is benzyl-aminopurine; "4PU-30" is the phenyl-urea cytokinin; and 'Ag⁺' is silver nitrate.

## Chemicals

N6-benzylaminopurine was obtained from Sigma Chemical Co (St. Louis, Mo, USA). N- (2-chloro-4pyridyl)-N′-phenylurea was kindly provided by Prof. K. Shudo, University of Tokyo, Japan. $AgNO_3$ was obtained from Merck.

## RESULTS

### Cotyledon area

First, we assayed cotyledon area (Fig. 2). Area was assayed 10 days after germination, several days longer than the duration of growth of the wild type in darkness (*Stoynova-Bakalova et al., 2004*), thereby allowing time for any effects on expansion to be completed. In darkness and without added nutrients, *A. thaliana* cotyledons expanded to a negligible extent (Fig. 2). Cytokinins are broadly classified as either purine-type or phenyl-urea-type and we tested one of each, benzyl-aminopurine (BA) and 4PU-30, respectively, both synthetic. Phenyl-urea cytokinins are of interest because of their chemical stability and independence of nucleotide metabolism (*Worakan et al., 2017*). Either of these cytokinins modestly promoted cotyledon expansion. As found previously

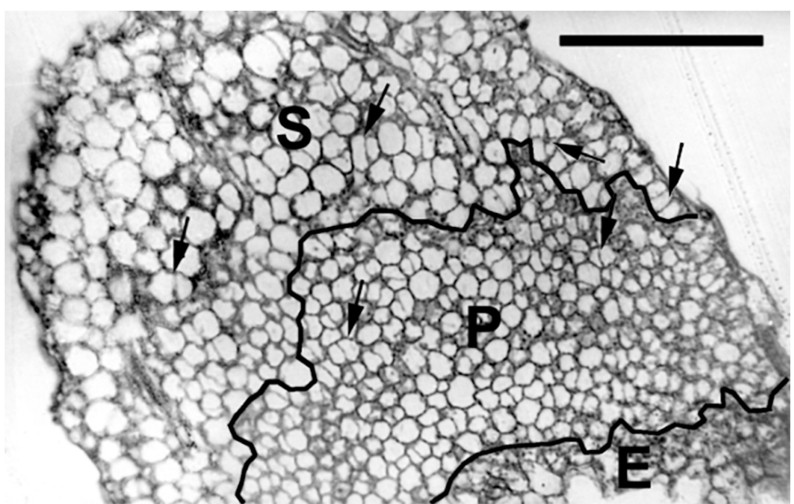

**Figure 3 Paradermal section of a cotyledon on the sixth day of development in darkness.** Figure shows general features of the cotyledons under the studied conditions. Section is of an *eti5* seedling, shown because this embedded cotyledon was unusually flat, allowing a large area to be imaged; a comparable image of the wild type has been published previously (Fig. 4A in *Stoynova-Bakalova et al., 2004*). Qualitatively, features of *eti5* and wild-type cotyledons appeared similar. Clusters were present in all cotyledon tissues (arrows). S, spongy mesophyll; P, palisade mesophyll; E, epidermis. Bar = 100 μm.           

for zucchini cotyledons (*Stoynova-Bakalova & Petrov, 2006*), 4PU-30 provoked a stronger effect than did benzyl-aminopurine. Expansion was also modestly stimulated when responses to ethylene were inhibited with silver nitrate, a widely used inhibitor of ethylene receptors (*Beyer, 1976*; *Kumar, Parvatam & Ravishankar, 2009*). Based on the above data, for *A. thaliana*, cotyledon growth is promoted by cytokinin and repressed by ethylene.

Notably, when cytokinin was added with silver nitrate to the wild type, expansion was promoted strongly. Insofar as silver nitrate might have off-target effects, either because of promiscuous binding by silver or because of extra nitrate, we carried out similar experiments in an ethylene-insensitive mutant, *eti5*. In many assays, this mutant scarcely responds to ethylene, even at a massive concentration (*Harpham et al., 1991*). Results with *eti5* were similar to those with silver nitrate (Fig. 2). Cotyledons of untreated *eti5* seedlings were larger than those of the wild type and when *eti5* seedlings were treated with the phenyl-urea cytokinin, cotyledon area was increased to a far greater extent than when that compound was given to the wild type. These results confirm and extend those published recently by *Cortleven et al. (2019)*, who abrogated ethylene responsiveness by using either silver nitrate or the ethylene-insensitive mutant, *ein2*.

## Cell division and expansion

To quantify growth and division of cotyledon cells, we assayed cell clusters in paradermal sections (*i.e.*, parallel to the blade; Figs. 1A and 3). Following germination, when the cells constituting the cotyledon of the dry seed divide, progeny cells form detectable clusters, an arrangement that permits divisions that happen after germination to be counted (Figs. 1A, 1B). Cell divisions in the cotyledons occur already at the second day of etiolated

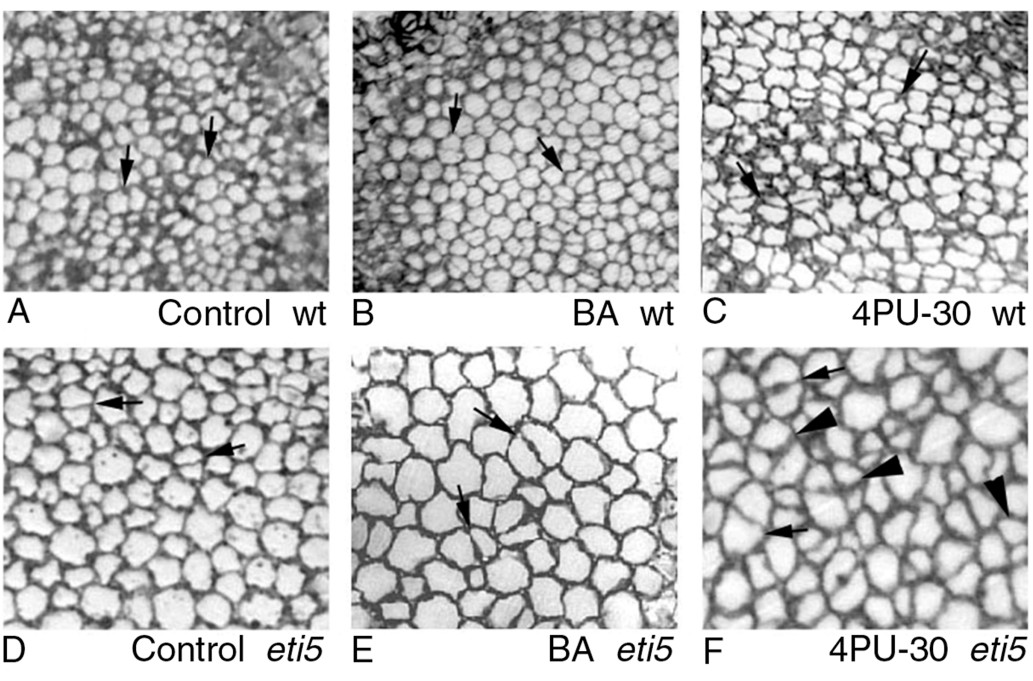

**Figure 4 Paradermal sections showing the central area of cotyledon blades in palisade tissue.** (A–C) Wild type. (D–F) *eti5*. Arrows indicate clusters of two progeny cells; arrowheads in (F) indicate clusters of three or four progeny cells. Bar = 50 μm.

development (Fig. 1A). By day 6, clusters with specific (linear) arrangement of three to four cells could be seen at the cotyledon margin, suggesting control over the plane of division (Fig. 3). In untreated cotyledons of both wild type and *eti5*, divisions occurred in all three tissues (epidermis, palisade, and spongy mesophyll). This is distinct from the epigeal cotyledons of zucchini and of some other species where the spongy mesophyll generally does not divide (*Stoynova-Bakalova, Petrov & Tsukaya, 2002*; *Stoynova-Bakalova, 2007*). Here, in both genotypes, spongy mesophyll cells were bigger than palisade cells, and were often surrounded by irregularly distributed intercellular spaces, sometimes even surpassing the dimensions of the neighboring cells (Figs. 3 and 4). In the palisade, intercellular air spaces were absent or at least too small to be seen. Because of distinct anatomy, we quantified division and expansion for palisade and spongy mesophyll cells separately.

As seen in the assayed sections, both large and small cells were able to divide (Figs. 3 and 4). In general, larger cells divided more frequently than smaller ones (*e.g.*, Fig. 4F, 4PU-30-treated *eti5* palisade cells). However, small cells did divide, even when adjacent to larger non-dividing cells (Figs. 3 and 4). Apparently, reaching a "critical volume" is not strictly necessary for a cell to divide and there is some uncoupling of the cell cycle from the control of cell size.

In general, the pattern of cotyledon area expansion (Fig. 2) was reflected by expansion of palisade cells (Fig. 5B) and of spongy mesophyll cells (Fig. 6B). Notably, untreated *eti5* had larger cotyledons and larger mesophyll cells than had the wild type, supporting the finding

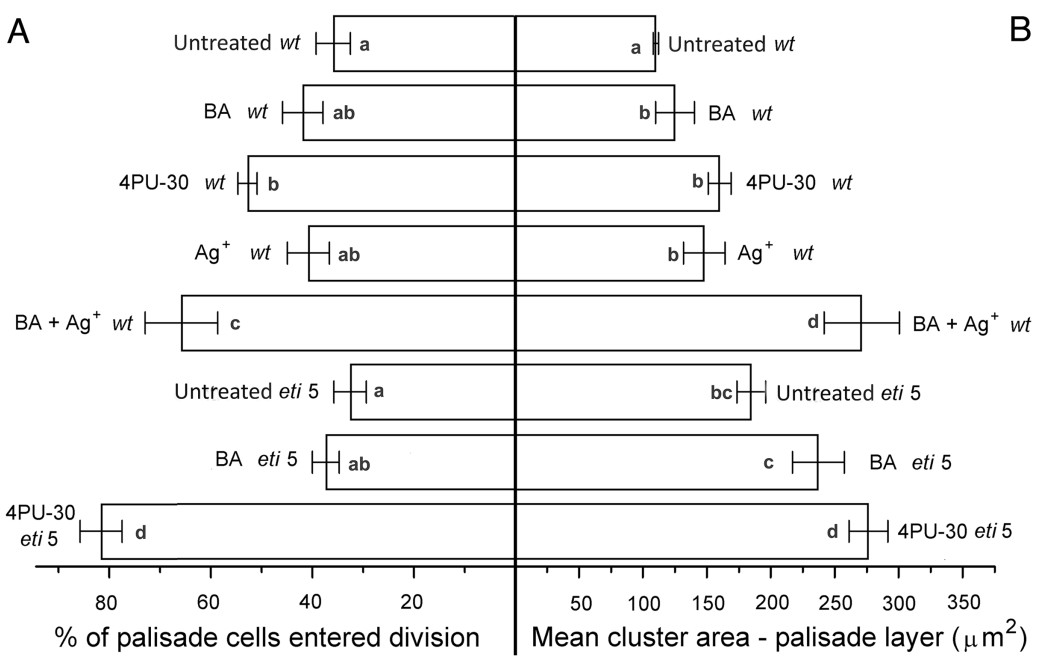

**Figure 5 Cell growth and division activity in cotyledon palisade mesophyll.** Bars represent mean (A) % of cells that have divided and (B) cluster area ± SE. Means on the left or on the right without a common superscript letter (a–d) differ ($P < 0.05$) as analyzed by two-way ANOVA and Fisher LSD test. At least 120 clusters were scored per treatment.

of a repressive role for ethylene in cotyledon expansion. When ethylene response was inhibited, cytokinin strongly promoted expansion of both palisade and mesophyll cells.

Division was affected similarly, with a significant increase in cells of both palisade and spongy mesophyll entering division when cytokinin was applied and ethylene response inhibited (Figs. 5A and 6A). In benzyl-adenine-treated *eti*5 cotyledons, cells usually entered division once, like those of similarly treated wild type (Fig. 4). By comparison, 4PU-30 stimulated cell division more strongly (Figs. 5A and 6A). The large "growth fraction" (*i.e.*, the number of dividing cells) in this treatment is due to multiple rounds of division for cells within a given cluster as well as more cells across the cotyledon entering the cell cycle. However, mesophyll cells (both palisade and spongy) in untreated *eti*5 tended to divide less than those of the wild type, suggesting a missing cell division promoter, but the differences were not significant. Nevertheless, ethylene repressed the ability of both types of cytokinin to promote division substantially, about as effectively as ethylene repressed that cytokinin's ability to promote expansion. Taken together, our results indicate that ethylene antagonizes cytokinin in both the cell division and expansion of dark-grown cotyledons.

## DISCUSSION

For the cotyledon, we show here that, on its own, cytokinin modestly promotes growth and division and endogenous ethylene modestly inhibits expansion; nevertheless, endogenous ethylene strikingly restrains the promoting activity of cytokinin on both expansion and division. We note that, in darkness, the nutrient reserves of the tiny *A. thaliana* seed will be

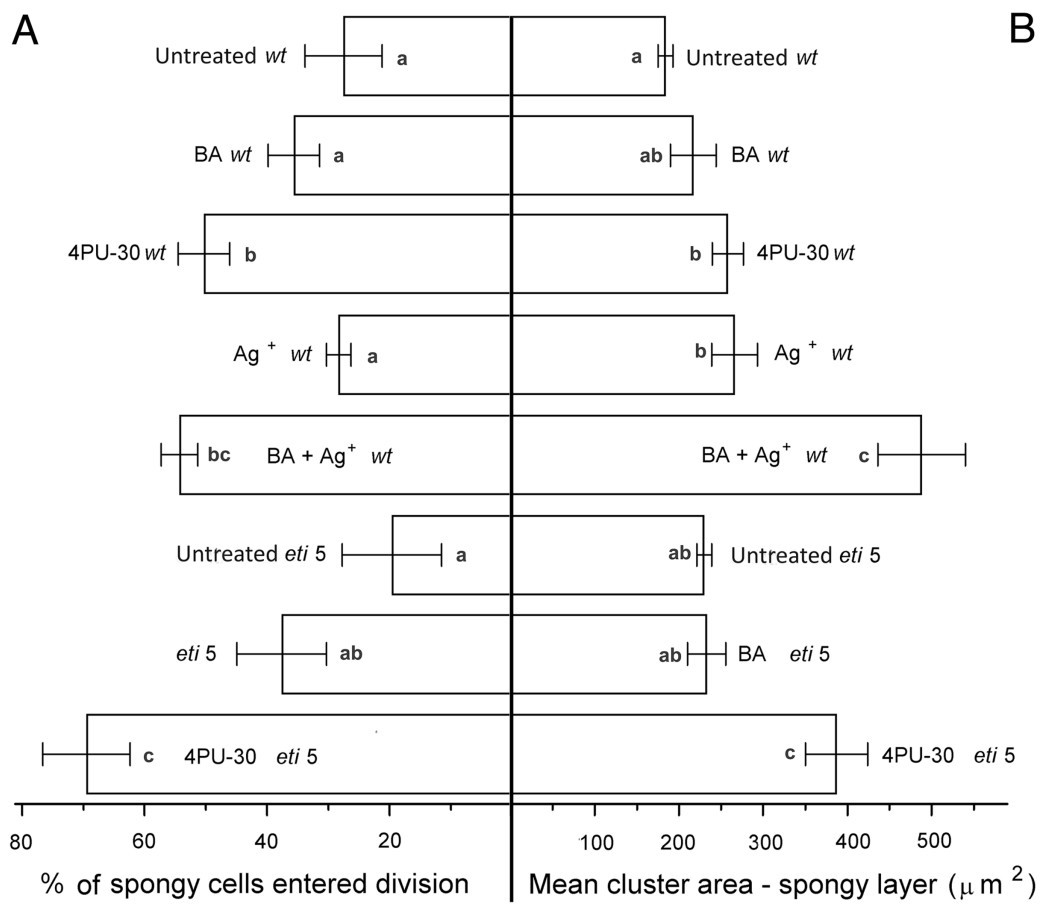

**Figure 6 Cell growth and division activity in cotyledon spongy mesophyll.** Bars represent mean (A) % of cells that have divided and (B) cluster area ± SE. Means on the left or on the right without a common superscript letter (a–c) differ ($P < 0.05$) as analyzed by two-way ANOVA and Fisher LSD test. At least 120 clusters were scored per treatment.

spent rapidly, suggesting that cytokinin-driven stimulation of growth will be even larger in the more replete cotyledons of other species. Our results contradict earlier statements claiming that cytokinin influences cotyledon growth independently of ethylene (*Cary, Liu & Howell, 1995*; *Vogel et al., 1998a*; *Smets et al., 2005*). Instead our results confirm the antagonistic relationship between the hormones for expansion reported by *Cortleven et al. (2019)* and extend it to cell division.

Here, to prevent ethylene responses, on one hand, we used silver nitrate (*Beyer, 1976*; *Kumar, Parvatam & Ravishankar, 2009*) and, on the other hand, the ethylene insensitive mutant, *eti5* (*Harpham et al., 1991*), with similar results in both cases. Although *eti5* was characterized as reflecting a single, semi-dominant allele (*Sanders et al., 1991*), the molecular nature of the mutation has to our knowledge never been identified. In etiolated seedlings, *eti5* is unresponsive to the highest concentration of ethylene tested (10,000 µL/L) for plumule angle, hypocotyl length, hypocotyl width, and root length (*Harpham et al., 1991*). In both of these respects (semi-dominance and sensitivity), *eti5* resembles *etr*, a mutant in a canonical ethylene receptor (*Bleecker et al., 1988*).

In addition to being strongly ethylene insensitive, *eti5* overproduces ethylene (*Harpham et al., 1991*) and also cytokinin (*Todorova et al., 2005*). Presumably, the overabundance of ethylene has little impact; however, in view of the reported excess of cytokinin, the stimulation of cotyledon growth and division seen here from treating *eti5* seedlings with cytokinin is notable. However, elevated cytokinin in *eti5* was seen in 3 to 5 week-old plants grown in the light; and in fact, cytokinin levels in older plants (7 weeks) were the same in *eti5* and in wild type (*Todorova et al., 2005*). To our knowledge, cytokinin levels have not been measured in etiolated *eti5* seedlings. Insofar as seedlings of *etr* have not been reported to over-produce cytokinin, this phenotype is not an inevitable consequence of ethylene irresponsiveness. In view of the strong ethylene insensitivity of *eti5* and its interaction with the cytokinin system, elucidating the molecular nature of the mutation is a worthwhile goal for future research.

In etiolated cotyledons of *A. thaliana*, suppressing ethylene action either chemically with silver nitrate or genetically with *eti5*, stimulated expansion. In this respect, etiolated cotyledons differ from light-grown leaves, insofar as ethylene insensitivity increases leaf area in neither *A. thaliana*, tobacco, nor petunia (*Tholen et al., 2004*). Similar to expansion in leaves, cell division in the etiolated cotyledons appears to be regulated weakly by baseline levels of ethylene. Division in the wild type was stimulated only very slightly in either kind of mesophyll tissue by silver nitrate; in *eti5*, untreated cotyledons tended to have fewer divisions than wild type, although the differences were not statistically significant. In general, cell division is rarely a clear target of ethylene. For example, in *A. thaliana* roots, where ethylene strongly inhibits cell expansion, the hormone sometimes has no effect on division (*Swarup et al., 2007*), at other times it shortens the meristem (*Thomann et al., 2009*; *Street et al., 2015*; *Zdarska et al., 2019*)—meristem shortening probably indicates action at the level of zonation rather than the cell cycle (*Baskin, 2013*)—and ethylene stimulates division in the quiescent centre (*Ortega-Martínez et al., 2007*). In shoots, ethylene modestly inhibits cell division in *A. thaliana* leaves (*Skirycz et al., 2011*) but stimulates division in cucumber hypocotyls (*Kazama et al., 2004*).

Here, for the cotyledon, when ethylene action was inhibited, cytokinin strongly stimulated cell division. Interestingly, in silver nitrate-treated wild type, division was stimulated more by the purine type than by the phenyl-urea type cytokinin, whereas the reverse was true for division in *eti5*. In view of cytokinins in leaves, and presumably cotyledons, being the subject of complex intersecting metabolic pathways (*Skalák et al., 2019*; *Li et al., 2021*; *Wu et al., 2021*), it is not surprising that various classes of cytokinin are more or less effective in silver nitrate-treated wild type compared to *eti5*.

Antagonism between ethylene and cytokinin, as seen here, is typical of how these hormones govern various developmental processes, including morphogenesis, organ senescence and resistance to pathogens (*Gan & Amasino, 1997*; *Hamant et al., 2002*; *Khaskheli et al., 2018*; *Kučerová et al., 2020*; *Veselova et al., 2021*). Interestingly, the exception appears to be root and hypocotyl growth, where at least under most conditions the hormones act cooperatively to inhibit expansion (*Deikman, 1997*; *Li et al., 2021*), apparently because the signal transduction machinery for the two hormones share components (*Liu et al., 2017*; *Zdarska et al., 2019*; *Bidon et al., 2020*). The current model of

signal transduction pathways for each of these hormones begin with two-component regulation; namely, a receptor histidine-kinase coupled to a response regulator (*Liu et al., 2017*; *Bidon et al., 2020*; *Park et al., 2021*; *Wu et al., 2021*). For both hormones, these components are encoded by small gene families and, for the cooperative growth inhibition in root and hypocotyl, components from one hormonal pathway apparently substitute for those of the other, *i.e.*, literal crosstalk (*Iqbal et al., 2017*; *Zdarska et al., 2019*). However, for the cotyledon, because cytokinin stimulated cotyledon growth when ethylene responsiveness was turned off, which is the opposite of what usually happens with the root or hypocotyl (*e.g.*, *Cary, Liu & Howell, 1995*), we suggest that, for cotyledon growth, the antagonistic action occurs *via* distinct, downstream targets.

The results presented here enrich the picture of cytokinin-ethylene interactions within an organ. Signal transduction pathways for these hormones have been well studied in hypocotyls and radicles, but comparatively less so in cotyledons. Hypocotyls and radicles, organs formed in the embryo, are popular for experiments because material is available soon after germination. The cotyledon, another organ arriving early, lacks the complex zonation of the root and has cells that expand and divide congruently, features providing a valuable comparative system to the root and hypocotyl for elucidating mechanisms of hormone action. Subsequent research at the molecular level should uncover whether the antagonism seen here between ethylene and cytokinin represents antagonistic crosstalk among the two-component regulators or instead features interactions well downstream.

## CONCLUSIONS

Here, for the cotyledon plate meristem of *A. thaliana*, ethylene is revealed as an inhibitor of expansion during etiolated cotyledon development and as an antagonist of cytokinin. Combining ethylene insensitivity with cytokinin treatment was synergistic, stimulating strongly both cell division and expansion. Evidently, ethylene actively represses the growth promoting influences of cytokinin. In these interactions, the cotyledon is distinguished from the root and hypocotyl. No doubt, cotyledon development involves other hormones, sensing and responding to the complex environment. The challenge is to understand how the plant converts the confusing crosstalk of the engineer to the integrated signaling crosstalk of the biologist. We believe that meeting this challenge can be aided by the epigeal cotyledon.

## ACKNOWLEDGEMENTS

We thank Prof. Michael Hall (University of Aberystwyth, Wales) for the gift of *eti5* seeds and for helpful discussions, Dr. Peter I. Petrov (Bulgarian Academy of Sciences) for technical assistance, and Prof. K. Shudo, University of Tokyo, Japan, for the gift of 4PU-30. Finally, we acknowledge the beneficial influence on the manuscript of several anonymous reviewers.

### Funding
The authors received no funding for this work.

### Competing Interests
Tobias I. Baskin is an Academic Editor for PeerJ. The authors declare that they have no other competing interests.

### Author Contributions
- Ekaterina Stoynova-Bakalova conceived and designed the experiments, performed the experiments, prepared figures and/or tables, authored or reviewed drafts of the article, and approved the final draft.
- Dimitar V. Bakalov analyzed the data, prepared figures and/or tables, and approved the final draft.
- Tobias I. Baskin analyzed the data, authored or reviewed drafts of the article, and approved the final draft.

### Data Availability
The raw data are available as a Supplemental File.

### Supplemental Information
Supplemental information for this article can be found online at http://dx.doi.org/10.7717/peerj.14315#supplemental-information.

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
