# Peer review of "Ethylene represses the promoting influence of cytokinin on cell division and expansion of cotyledons in etiolated Arabidopsis thaliana seedlings"

_PeerJ, doi:10.7717/peerj.14315_

## Round 0.1 · original submission · Major Revisions

The manuscript has been correctly evaluated but requires some attention regarding details and precision on the set-up, the clarity of the explanations, some rephrasing or re-arrangement of the text. Please take carefully into account the comments before resubmitting your work.

Reviewer 1 ·

Basic reporting

In general, the manuscript is written in good English.
However, the authors should revise the text as there are several typos and ambiguities.
For example:
line 55: Wang et al., 2021 please change to Wang et al., 2019
line 105: starting from this line onwards Columbia background please change to Col-0 background
line 156: depends partially by ethylene please change to depends partially on ethylene
line 181: starting from this line onwards – change Fig’s 3, 4 to Figs 3, 4 etc.
line 219: semi-dominant gene please change to semi-dominant mutation
line 250: ethylene was inhibited please change to ethylene action was inhibited


In general, literature is referenced appropriately. However, it is somewhat surprising that on lines 62 and 63, no direct reference on action of cytokinin directly in Arabidopsis is given though several reports in this area are available, e.g. Skalak et al., Plant J., 2019, 805-824.


The manuscript obeys to rules valid for article structure, figures and tables.


The manuscript is self-contained.

Experimental design

The manuscript falls into Aims and Scope of PeerJ.

Questions are clearly defined and relevant. Nevertheless, they just scratch the surface of the topic. A deeper insight into the topic could be obtained if an appropriate set of mutants were analyzed and data obtained evaluated. Material and methods are described in a way allowing for replication of the experiments.

However, there is a problem with a mutant (eti5) selected to study cytokinin action in absence of ethylene signaling / ethylene sensitivity. Authors state that the molecular nature of the mutation is unknown. Further, eti5 overproduces ethylene and cytokinin. So what makes eti5 a good candidate to investigate cytokinin-ethylene cross-talk in regulation of cell division and growth? Authors reason that cytokinin is not necessarily increased at the developmental stage under investigation. However, they do not prove it by determination of cytokinin content. Why etr1 was not selected for the work?

Validity of the findings

All underlying data are presented and statistically evaluated.

Conclusions are well stated, relevant to the research question and supported by the results.

Additional comments

The manuscript is reminding an importance of nature of an organ in determining outcomes of hormonal cross-talks which underlie regulation of cell division and expansion which in turn determine size of the organ in question. The experimental system employed is interesting and used only rarely. The “phenotyping” approach could bring much deeper insight into the topic if data were obtained on an appropriate set of mutants and analyzed.

Reviewer 2 ·

Basic reporting

English language in this manuscript needs a little improvement. Some advanced vocabulary can be replaced to make the manuscript be more acceptable to audiences.
The objective of this study is not clearly described in the introduction.
More details should be included in the Materials and Methods.
Results in this manuscript need to be re-organized. Some parts in present edition should be included in Materials and Methods, some parts should be in Discussion. More findings of your experiment should be included in Results.
Legends of the figures need to be improved. Detailed advice can be found in following comments.

Our goal should be making international audience working in different areas can clearly understand what was done in your research. I believe with some improvement in language and organization, more details added to Materials and Methods, and Results, we can achieve this goal.

Experimental design

The format of DOIs in references needs to be unified.
Line 108-110: How much volume of these solutions were added? How long were cotyledons being treated? Please describe all treatments in Materials and Methods.
What are the effects of AgNO3, and 4PU-30? You may add them to the Introduction.
Line 127: What is the definition of daughter cell and cluster?
Line 129: What is the experimental design?

Validity of the findings

Line 152: ‘As found previously for zucchini cotyledons (Stoynova-Bakalova & Petrov, 2006)’ should be a part of discussion instead of results, similar problems were found in lines 161, 165, and 178.
Line 155: ‘responses to ethylene were inhibited with silver nitrate, a widely used inhibitor of ethylene receptors (Beyer, 1976; Kumar, Parvatam & Ravishankar, 2009)’ should be included in introduction.
Line 153: Is the ‘benzyl-aminopurine’ here the same as BA used in your study? If yes, please use abbreviation here. Also, you should mention BA used in your treatment is a synthetic cytokinin.
Line 156: ‘For A. thaliana, cotyledon growth apparently depends partially by ethylene.’ How did you get this conclusion? Your conclusion should not come from references. In Results, you should concisely describe what was found in your experiment. Don’t interpret or explain your data.
Line 158-160: This part should be explained in Materials and Methods. In addition, why there is no BA and Ag+ combined treatment on eti5?
Figure 4: You may name the six pictures as a, b, c, d, e, and f. Arrowheads in Figure 4a-4c imply daughter cells, arrowheads in Figure 4d-4f imply clusters of 3 or 4 daughter cells.
Line 185: (e.g., Fig. 4, 4PU-30-treated eti5 palisade cells) can be changed as Fig. 4f based on above nomenclature.
Line 191-192: ‘implying a repressive role for ethylene in cotyledon expansion’ should be included in discussion part.
Line 197: ‘By comparison, 4PU-30 stimulated cell division more strongly.’ Based on which figure did you get this conclusion?
Line 198-205: This part should be included in discussion.
More description about Figure 5 and Figure 6 should be included in Results.
Line 228: ‘However, elevated cytokinin in eti5 was seen in 3 to 5 week-old plants grown in the light; and in fact, cytokinin levels in older plants (7 weeks) were the same in eti5 and in wild type.’ Please add reference.
Conclusions: The first paragraph should not be included in conclusions. In the second paragraph, however, only the first sentence can be treated as conclusion. You may rewrite the conclusions.

---

## Round 0.2 · Major Revisions

Although one reviewer was ok for publishing your manuscript, there are still changes that are necessary to accept it. Please check the reviewers' comments and make the appropriate modifications before resubmission.

Reviewer 1 ·

Basic reporting

no comment

Experimental design

no comment

Validity of the findings

no comment

Reviewer 2 ·

Basic reporting

no comment

Experimental design

no comment

Validity of the findings

no comment

Additional comments

I understand including references/interpretative statements in results can help some audiences. However, exact experimental outcome description (result) is still missing in the revised manuscript. Since ‘Conclusions’ were deleted, please add ‘Conclusions’ to your manuscript. In the instructions for authors of PeerJ, conclusions is required (https://peerj.com/about/author-instructions/).
If this manuscript is writen as a short communication instead of a full-length paper, I recommend authors to submit it to a journal with a short communication category instead of PeerJ.

Reviewer 3 ·

Basic reporting

The manuscript by Stoynova-Bakalova and colleagues explores an interesting mechanism that delves into the cytokines-ethylene cross-talk during cotyledon development, focusing on cell division and elongation that determines the organ size. These results have been validated at genetic and pharmacological levels. Additionally, the authors have made the modifications -or made the clarifications, suggested by previous reviewers on the manuscript.
Overall, the approach described in the manuscript is interesting. Although the results are relevant the manuscript has some flaws presented below that should be addressed.

Introduction
Lines 68 and 78 sound repetitive. Please, select another connector.

Materials and methods
In general, the procedures are clear and correct but some points need to be clarified. For example:
1. The author states: “[…] three independent experiments”, “[…] cells were counted in at least 120 clusters from 10 to 12 cotyledons.”. It would be necessary to indicate how many plants were used in each replicate. Usually, 15 seedlings per replicate (15 to 30 cotyledons). Please, clarify that.
2. In “Chemicals”. Please, include the references for the reagents.

Results
1. Fig. 1. Please, include a pictogram describing/identifying cell clusters and cotyledon zones. This could enhance the data interpretation.
2. To demonstrate the findings shown in Fig. 3., a paradermal section of the cotyledon of a wild type of plant is needed.

The discussion and Bibliography are correct, and the authors have made the corrections according to the reviewers.

Experimental design

The experimental procedures are an adequate approach to the study, allowing to address the question about the cytokines-ethylene cross-talk during cotyledon development. Nevertheless, the manuscript has some flaws presented below that should be addressed.

According to reviewer 1, and despite the author's explanations a treatment with BA+AgNO3 could be positive to validate the findings in eti5 mutant and wild type (e.g. Treated with BA+AgNO3 or eti5 treated with BA).
The manuscript falls into the Aims and Scope of PeerJ.

Validity of the findings

The findings are statistically evaluated, and the raw data is correct. Nevertheless, the authors need to clarify the number of seedlings analyzed on each replicate.
Conclusions are clear and relevant to the biological question and backed by the results.

Additional comments

The manuscript is written correctly, and its structure allows the progression in the explanation of the biological question of the research. Additionally, a deep study of the molecular/cellular mechanisms that are regulating cell elongation and division in cotyledons will be necessary for the future. Nevertheless, the cellular approach used to understand the cross-talk between cytokines and ethylene is correct and sufficient.

---

## Round 0.3 · accepted · Accept

Given the conditions you've explained in the rebuttal letter and response to reviewers' comments, I've assessed the revision myself and I'm satisfied with the current version.